# Children Born with Congenital Heart Defects and Growth Restriction at Birth: A Systematic Review and Meta-Analysis

**DOI:** 10.3390/ijerph17093056

**Published:** 2020-04-28

**Authors:** Ali Ghanchi, Neil Derridj, Damien Bonnet, Nathalie Bertille, Laurent J. Salomon, Babak Khoshnood

**Affiliations:** 1Université de Paris, CRESS, INSERM, INRA, F-75004 Paris, France; 2Service d’Obstétrique—Maternité, chirurgie médecine et imagerie fœtales, APHP, Hôpital Necker Enfants Malades, F-75015 Paris, France; 3Department of Pediatric Cardiology, M3C-Necker, APHP, Hôpital Necker-Enfants Malades, F-75015 Paris, France

**Keywords:** congenital heart defects, small for gestational age, systematic review, meta-analysis, population-based study

## Abstract

Newborns with congenital heart defects tend to have a higher risk of growth restriction, which can be an independent risk factor for adverse outcomes. To date, a systematic review of the relation between congenital heart defects (CHD) and growth restriction at birth, most commonly estimated by its imperfect proxy small for gestational age (SGA), has not been conducted. Objective: To conduct a systematic review and meta-analysis to estimate the proportion of children born with CHD that are small for gestational age (SGA). Methods: The search was carried out from inception until 31 March 2019 on Pubmed and Embase databases. Studies were screened and selected by two independent reviewers who used a predetermined data extraction form to obtain data from studies. Bias was assessed using the Critical Appraisal Skills Programme (CASP) checklist. The database search identified 1783 potentially relevant publications, of which 38 studies were found to be relevant to the study question. A total of 18 studies contained sufficient data for a meta-analysis, which was done using a random effects model. Results: The pooled proportion of SGA in all CHD was 20% (95% CI 16%–24%) and 14% (95% CI 13%–16%) for isolated CHD. Proportion of SGA varied across different CHD ranging from 30% (95% CI 24%–37%) for Tetralogy of Fallot to 12% (95% CI 7%–18%) for isolated atrial septal defect. The majority of studies included in the meta-analysis were population-based studies published after 2010. Conclusion: The overall proportion of SGA in all CHD was 2-fold higher whereas for isolated CHD, 1.4-fold higher than the expected proportion in the general population. Although few studies have looked at SGA for different subtypes of CHD, the observed variability of SGA by subtypes suggests that growth restriction at birth in CHD may be due to different pathophysiological mechanisms.

## 1. Introduction 

Congenital heart defects (CHD) are the most common group of congenital anomalies with a live birth prevalence of 8.2 per 1000 births in Europe [1]. Despite considerable progress in medical and surgical management of CHD, they remain the most important cause of infant death by malformation. One study suggested that there were approximately 260,000 deaths due to CHD in 2017 [2]. However, the survival rate is much higher in high resource countries and a recent review found that 85% of children with CHD reach adulthood [3].

Growth restriction at birth, often measured by its imperfect proxy small for gestational age is an important risk factor for perinatal mortality, morbidity, and long-term adverse outcomes, including an increased risk of diabetes, hypertension, and cardiovascular disease later in life.

Therefore, growth restriction in a newborn with a CHD may represent a “double jeopardy” with risks related to CHD combined with those associated with growth restriction. Moreover, differences in the proportion of CHD subtypes with growth restriction may provide clues about possible pathophysiological mechanisms of the relation between growth restriction and CHD. 

To date, no systematic review of the relation between CHD and growth restriction at birth has been conducted. The objective of our study was to conduct a systematic review and meta-analysis of the relation between growth restriction at birth and CHD.

## 2. Methods

This study is reported in accordance to Preferred Reporting Items for Systematic Review and Meta-analyses (PRISMA) guidelines [4]. The review protocol was registered on the PROSPERO: International Prospective Register of Systematic Reviews website [5]. As data sources originated from previously published studies in the public domain, ethical approval for this study was not requested [6]. 

### 2.1. Search Strategy

A comprehensive literature search was carried out on Pubmed/Medline and Embase databases with the assistance of a specialized documentalist. Medical Subject Headings (MeSH)/Medical Embase Medical Headings (EMTREE) and keywords that included different synonyms for CHD, CHD subtypes, small for gestational age (SGA), fetal growth restriction (FGR)/intrauterine growth retardation (IUGR) and low birth weight were combined together using Boolean operators. The search was carried out from inception until 31/03/2019 and no language preferences were applied. A manual search of references in included articles was carried out to complete the search. 

### 2.2. Study Selection 

Titles and abstracts of retrieved studies were screened independently by two blinded reviewers (AG and ND) using Rayaan web application [7]. Excluded articles were about CHD and low birth weight only, conference abstracts, CHD and single umbilical artery, absence of SGA data, matched case control studies, use of estimated fetal weight from ultrasound data, and SGA outcomes in the offspring of women born with CHD.

### 2.3. Data Extraction 

A predetermined data extraction form was designed and used independently by the two reviewers (AG and ND). Extracted data for each study included study characteristics, object of study, SGA outcomes, data sources, exclusion criteria, and SGA proportions. Authors of studies were contacted to request further information or clarification of results.

### 2.4. Evaluation of Bias

The Critical Appraisal Skills Programme (CASP) cohort study checklist evaluated the risk of bias in studies included in this review [8]. The checklist contains 12 questions divided into three sections that enable a structured approach to finding evidence, determine possible sources of bias, and evaluate internal and external validity of each study. We adapted this checklist to our study question paying particular attention to selection and measurement biases. 

Throughout the entire process (article selection, data extraction, and evaluation of bias) discrepancies were resolved through end result discussion. Any further disagreements between the two reviewers (AG and ND) were resolved by a third reviewer (BK). 

### 2.5. Definitions

CHD was defined as children born with structural heart defect and excluded patent ductus arteriosus, cardiac tumors, cardiomyopathies, and arrhythmias. Isolated CHD was defined as CHD not associated with chromosomal anomalies, malformations from other systems or syndromes. Due to data availability, we used SGA as an imperfect measure of growth restriction at birth. We used the consensual definition of SGA, defined as birthweight <10th percentile according to gestational age and compared to a standard population [9]. Studies were grouped according to birthweight percentile cut-off rather than labels assigned by the different authors. 

### 2.6. Statistical Analysis

A meta-analysis of pooled proportions (with their 95% confidence intervals) was carried out using a random effects model with inverse variance weighting, using the Simonian and Laird method [10,11]. Freeman–Tukey double arcsine transformation was used to limit the effects of over-weighting caused by studies with a variance close to zero for estimating the confidence intervals for the pooled estimate [10,11]. The I^2^ statistic assessed statistical heterogeneity between groups. Principal analysis concerned all/isolated CHD using the SGA defined using the 10th percentile cutoff threshold. Additional analyses were conducted for CHD subtypes and for severe SGA using the 3rd percentile. Sensitivity analysis was carried by restricting the analysis to only population-based studies. The meta-analysis was performed using STATA 12.1 software (StataCorp LP., College Station, TX, USA). We considered *p*-values < 0.05 as statistically significant.

## 3. Results

The database search identified 1783 potentially relevant publications of which 72 articles were assessed for eligibility. An additional two studies were found through hand searching of reference lists [12,13]. In total 38 studies were found to be relevant to the study question of which 18 citations contained sufficient data for a meta-analysis (Figure 1). 

### 3.1. Study Characteristics

Characteristics of the studies according to year of publication, country and objective of the study are shown in Table 1. Publication years ranged from 1972 to 2018 and 23 (60.5%) studies were published between 2010 and 2019. Sample sizes of patients with CHD ranged from 16 to 99,786. Twenty-six studies (68.5%) were based on US cohorts. The reference populations varied greatly based on geographical location and the year of study. Overall, 19 different reference populations were cited. The most frequent was growth curve by Alexander et al., which was used in six American studies while eight (21%) studies did not state which reference population was used.

Of the 38 studies included in the systematic review, 22 (57.9%) used birthweight <10th percentile) for definition of SGA; 17 (44.7%) studies were designed specifically to study SGA and CHD as their primary objective. Six studies (27.2%) did not report explicitly the use of gestational age or a reference population in their definition of SGA, whereas six studies (27.2%) studies considered gender in addition to gestational age in the definition of SGA (Table 1). Three (7.9%) studies used the term FGR even though the actual outcome was SGA. 

Twenty-three (60.5%) studies comprised all CHD and 10 (26.3%) isolated CHD only. In addition, 12 specific subgroups were studied with the majority of studies on hypoplastic left heart syndrome (HLHS) and Tetralogy of Fallot (ToF) (10 publications). 

### 3.2. Proportion of SGA in All CHD, Isolated CHD, and Subgroups Reported by Individual Studies 

As shown in Table 2, the proportions of SGA in all, isolated, and subgroups of CHD varied greatly across the studies in the systematic review. It was found that four (10.5%) studies on isolated CHD reported same proportion of SGA i.e., 15%. The proportion of SGA varied between 3% and 37% for HLHS 8% and 67% for ToF and 10% and 40% for ventricular septal defects and 5% and 57% for coarctation of the aorta (CoAo).

Some studies were restricted to preterm births or very low birth weight infants even though by far most studies included all gestational ages. Certain studies included a selected set of newborns with CHD, e.g., those operated for critical CHD. Only one study examined SGA for isolated CHD subgroups [14]. 

### 3.3. Evaluation of Bias

Studies were evaluated for bias using a modified CASP checklist. Yu et al. was omitted because we could not obtain the full article [15]. All studies addressed a clearly focused issue, however the quality of studies regarding other criteria in the checklist varied greatly. In particular, most studies were to some extent subject to selection and measurement bias, especially with regards to diagnosis of CHD using a validated diagnostic method. 

Few studies took into consideration the effects of confounding factors (e.g., parity, ethnicity, maternal disease, maternal smoking, etc.). Four studies were found to have a lower risk of bias [15,16,17,18], whereas five others were deemed to have a higher risk of bias [12,19,20,21,22]. Confidence intervals (CI) for SGA proportions were not provided in any study. Notwithstanding differences in geographic locations and reference populations, external validity criterion was met for most studies as they were population-based. 

### 3.4. Meta-Analysis

Of the 38 articles in the systematic review, we used 18 (47.4%) in the meta-analysis. The reasons for excluding studies from the meta-analysis are detailed in Figure 1. These included studies of low birth weight and preterm newborns only, unclear definition or of CHD subgroups included, absence of data on birth weight or clear definition of SGA, and studies limited to one gender only.

The pooled proportion of SGA in all CHD was 20% (95% CI 16–24%) and for isolated CHD 14% (95% CI 13–16%) (Figure 2). Limiting the meta-analysis only to population-based studies did not change the results appreciably. Based on two studies that used the 3rd percentile, the proportion of severe SGA for all CHD was 6% (95% CI 6–7%). 

Table 3 illustrates the results of meta-analysis for subgroups of CHD. Genetic and other anomalies were not explicitly excluded in the studies reporting on subgroups of CHD. Pooled proportion of SGA was 30% for ToF, 21% for HLHS, and 17% for transposition of great vessels (TGV). The proportion of SGA was lowest for isolated atrial septal defects (ASD) with a proportion of 12%.

**Table 2 ijerph-17-03056-t002:** Summary of key characteristics of individual studies.

Author	Country	Definition of SGA	CHD	CHD (*n*)	SGA (%)
Archer (2011) [23]	USA	<10th P° according to GA, maternal race, gender, and type of gestation	All	99,786	21
Bain (2014) [24]	USA	<10th P° according to GA, gender, race	All	98,523	24
Calderon (2018) [25]	France	<10th P° according to GA and gender	All	419	14
Cedergren (2006) * [26]	Sweden	<2SD below mean birth weight according to GA	All	6346	7
			Isolated	5338	6
Chu (2015) [27]	USA	ICD?	All	28,806	6
Cnota (2013) [28]	USA	<10th P° according to GA, gender, race	HLHS	33	No data
Joelsson (2001) [29]	Sweden	Not stated	PAIVS	84	14
El Hassan (2008) [30]	USA	ICD	HLHS	5720	3
Fisher (2015) [31]	USA	Not stated	All	235,643	43
Gelehrter (2011) * [32]	USA	<3rd P° according to GA	HLHS	52	37
Jacobs (2003) * [33]	China	<-2 z score from normal mean for age and gender	Isolated	454	15
			PA	18	11
			ToF	63	24
			TGV	12	16
			CoAo	20	20
			VSD	86	12
			ASD	31	23
			PS	52	11
Jones (2015) * [20]	USA	<10th P° according to GA and gender	HLHS	16	31
Josefsson (2011) [34]	Sweden	<-2 SD of the mean birthweight for gestational length	All	2216	31
Karr (1992) [35]	USA	Not stated	ToF	125	21
Kernell (2014) [36]	Sweden	<-2 SD of the mean birthweight for gestational length	All	2689	21
Khoury (1988) * [12]	USA	<10th P° according to GA, race and gender	All	3669	28
			HLHS	91	23
			CAT	34	24
			ToF	110	33
			TGV	167	17
			CoAo	139	28
			VSD	833	27
			ASD	409	30
			i.ASD	26	11
			AVSD	103	28
Kramer (1990) * [37]	West Germany	<10P°	Isolated	843	15
			ToF	81	26
			TGV	60	15
			AS	45	8
			CoAo	69	13
			VSD	236	13
			ASD	70	17
Levin (1975) [38]	USA	Not stated	All	37	43
			VSD	5	40
			AoA	3	70
Levy (1978) * [39]	USA	<2SD below mean birth weight of control group	All	2178	6
			HLHS	163	6
			TA	64	5
			TAPVR	58	3
			ToF	156	7
			TGV	217	2
			AS	43	2
			CoAo	136	6
			VSD	313	10
			ASD	59	8
			AVSD	107	8
			PS	81	5
			PAIVS	64	6
Li (2009) [21]	China	Not stated	All	274	5
Lupo (2011) [40]	USA	<10th P° according to GA and gender	Ebstein	175	19
Malik (2007) * [16]	USA	<10th P° according to GA and gender	Isolated	3395	15
Nembhard (2009) * [41]	USA	<10th P° using race specific growth curve	All	9645	19
			HLHS	283	23
			CAT	112	25
			ToF	602	26
			Ebstein	61	15
			TGV	472	20
			CoAo	592	20
			VSD	5528	17
			ASD	467	28
Nembhard (2007) * [17]	USA	<10th P° using race specific growth curve	All	12,964	16
			Isolated	10,870	13
Oyarzún (2018) [22]	Chile	Not stated	Isolated	46	26
Pappas (2012) [42]	USA	<10th P°	All	110	27
Polito (2013) [43]	Italy	<3rd P°	All	70	17
Reynolds (1972) * [13]	USA	<10th P° according to GA	All	433	14
			AS	21	38
Rosenthal (1991) * [14]	USA	<10th P° according to GA	Isolated	1299	12
			HLHS	96	20
			CAT	113	18
			ToF	119	7
			Ebstein	57	5
			TGV	103	10
			CoAo	470	11
			VSD	130	12
			ASD	44	18
			PS	167	14
Sochet (2013) [44]	USA	<10th P° according to GA	All	230	25
Steurer (2018) * [45]	USA	<10th P° according to GA and sex	Isolated	6863	16
Story (2015) * [46]	UK	<10th P°	Isolated	308	16
Swenson (2012) * [47]	USA	<10th P°	All	753	21
			HLHS	261	19
			TA	38	16
			CAT	28	21
			DROV	54	24
			TAPVR	35	26
			ToF	70	36
			TGV	181	13
			IAA	44	36
			AVSD	25	32
Wallenstein (2012) * [18]	USA	<10th P°	All	193	24
			Isolated	129	15
Wei (2015) [48]	USA	Size < 10th P°	All	74	51
			HLHS	11	30
			ToF	12	70
			Ebstein	4	50
			CoAo	7	57
			VSD	6	17
			PAIVS	5	60
Williams (2010) * [49]	USA	<10th P° according to GA	HLHS	606	20
			TA	114	30
			AVSD	148	25
			PAIVS	102	25
Wollins (2001) * [19]	USA	<10th P° according to sex and GA	CoAo	181	12
Yu (2014) [15]	China	Not stated	All	477	11

Legend: * included in meta-analysis. § Not a population-based study. σ SGA 1st aim of study. SGA—small for gestational age; CHD—congenital heart defect; HLHS—hypoplastic left heart syndrome; ToF—Tetralogy of Fallot; VSD—ventricular septal defect; CoAo—coarctation of the aorta; TGV—transposition of great vessels; AVSD—atrioventricular septal defect; ASD—atrial septal defect; i.ASD—isolated atrial septal defect; TA—tricuspid atresia; CAT—common truncus arteriosus; PAIVS—pulmonary atresia intact ventricular septum; TAPVR—total anomalous pulmonary venous return; DORV—double outlet right ventricle; IAA—interrupted aortic arch; AoA—aortic atresia; PS—pulmonary stenosis; AS—aortic stenosis; P°—percentile; GA—gestational age; SD—standard deviation; ICD—international classification of diseases.

## 4. Discussion

### 4.1. Main Findings and Interpretations

This systematic review and meta-analysis found 38 articles that studied the association between SGA and CHD. The pooled proportion of SGA for all CHD was 20% and for isolated CHD 14%. Given the definition of SGA as the 10th percentile, these results suggest that overall, newborns with CHD have a two-fold greater risk of SGA compared to its theoretical value and those with isolated CHD a 1.4-fold higher risk of SGA. Estimates of SGA in the general population in developed countries are also considerably lower than the pooled proportions in our meta-analysis [50,51]. There was a great deal of variability in the proportion of SGA for different CHD. Tetralogy of Fallot had the highest proportion of SGA whereas isolated ASD had the lowest proportion of SGA. The range of SGA proportions across studies was highly variable for CHD, isolated CHD, or given subgroups of CHD in the 38 studies included in the systematic review. However, this variability decreased substantially for the 18 studies included in the meta-analysis.

Overall, approximately 20%–30% of CHD are due to known chromosomal, genetic, or other anomalies [52,53]. Some of these anomalies, e.g., Down Syndrome, Turner Syndrome may in turn be associated with growth restrictions. Indeed, isolated CHD had a substantially lower proportion of SGA. The issue of associated anomalies complicates the interpretation of differences in subgroups of CHD as they may be more (ToF) or less (HLHS or CoA) associated with other anomalies.

The higher proportion of SGA in newborns with CHD may be caused either by the CHD itself and/or by a common etiological factor (maternal, fetal, placental) that can cause both CHD and growth restriction [12,16,52,54].

With regards to the theory that CHD causes SGA, a number of authors suggest that alterations in fetal hemodynamics and oxygen saturation due to CHD are the root cause of this association [12,14,16,51]. Differences in SGA proportions according to CHD subtypes that we identified in this review support this hypothesis with the proportions of SGA varying from 22% for CoA to 12% in isolated ASD. Wallenstein et al. hypothesized that reduced ventricular function decreases cardiac output resulting in stunted fetal growth [18]. Our findings of increased SGA in HLHS (21%) are consistent with this mechanism. Story et al. maintained that decreased oxygenation in the aortic arch reduces cerebral perfusion and thus causes SGA [46]. Our findings of increased proportions of SGA in transposition of great arteries (TGA) (17%) may be at least in part explained by this mechanism. Sun et al. also found that decreased oxygen consumption is associated with smaller brain sizes in children with CHD [55].

Several authors have hypothesized that the association between SGA and CHD is caused by one or more common etiological factors (maternal, placental, fetal, and/or environmental) that result in both CHD and SGA [20,54]. Malik et al. have proposed that smoking may contribute to a common etiological pathway for CHD and SGA [56]. Although 33 studies (86%) included in our review provided data on maternal smoking only four (11%) took this into consideration in their statistical analysis [14,18,19,26]. Cedergren and Kallen theorized that disturbed placentation caused by abnormal trophoblastic growth in early pregnancy results in both SGA and CHD [26]. While, Jones et al. argued that placental insufficiency is the common causal pathway for HLHS [20]. They asserted that placental insufficiency reduces angiogenesis and villous tree maturation of the placenta, thereby reducing the surface area for gaseous and nutritional exchanges. As a result, SGA is induced directly and indirectly by nutritional deficiency. Their observations of increased placental leptin secretion led them to speculate that a predisposition for HLHS is the result of some kind of compensatory mechanism. Nevertheless, the effect of leptin in myocardial hypertrophy is debatable in the literature [57].

In addition to the two possible physiopathological mechanisms previously discussed, Spiers et al. proposed another, even if a minority position, hypothesis in the literature [12,14,46,58] According to Spiers et al., early FGR during cardiogenesis may result in CHD; in other words, SGA may be the cause of CHD [46,58]. Despite the fact that early FGR is very difficult to diagnose, five authors in this review made reference to this theory to account for the genetic anomalies and syndromes that are associated with CHD. They used this theory to explain that an intrinsic disturbance in fetal growth could provide a predisposition for CHD. However, to our knowledge little evidence exists to corroborate this theory.

In general, our results raise several questions about the possible underlying mechanisms of the association between SGA and CHD. Few studies were designed to examine this association specifically or to investigate different mechanisms that may explain the association between CHD and SGA. Moreover, the roles of confounding, intermediate (mediating) variables, and possible interactions in the causal pathway(s) between CHD and SGA have not been adequately studied. For example, the role of maternal age, if any, is unclear. While it is well known that maternal age (and parity) are associated with SGA, whether or not maternal age (or parity) in and of itself are risk factors for CHD is not known. Previous studies have provided conflicting results about the possible association between maternal age and CHD even if maternal age is known to be associated with SGA [3,59,60,61,62].

The genetic mechanisms potentially related to the association between CHD and SGA appear to be the result of complex, multifactorial interactions between genetics, epigenetics, and the environment that are poorly understood [61,62,63]. Certain specific isolated CHD subtypes may be caused by point mutations to transcription factors of specific genes (e.g., IRX4 results in VSD) that affect cardiogenesis. The expression of genes either directly (through methylation or other mechanisms) or indirectly via environmental exposure has been associated with CHD. DNA methylation was one of the first epigenetic mechanisms to be associated with CHD e.g., aberrant methylation of NKX2–5 and HAND1 genes has been observed to result in TOF [62]. A hypomethylative state of certain maternal genes may result in CHD being inherited in the offspring [64,65]. Monteagudo-sanchez et al. found that aberrant methylation of placental genes resulted in FGR although to our knowledge no study has yet to investigate hypomethylation of genes that cause both CHD and SGA [64]. Alternatively, chromatin remodeling and histone modification may also result in CHD epigenesis e.g., inactivation of deacetylases 5 and 9 are a feature of lethal VSD [61,62]. Small non-coding RNA may also contribute to the epigenetics of CHD with recent studies indicating that they are highly susceptible to environmental exposures e.g., cigarette smoking [60,65]. Similarly, through the same physiopathological pathways, maternal diabetes and obesity may induce CHD [61]. However, no study has specifically investigated the role of genetics or epigenetics in the association between SGA and CHD.

Another unresolved issue concerns the role of multiple pregnancies and its possible effect in the association between CHD and SGA. Although, Gijtenbeek et al. found in a systematic review that there is more CHD in twin pregnancies, which in turn are known to have higher rates of SGA [66]. Consequently, the link between multiple pregnancy and advanced maternal on CHD and SGA is unclear because to our knowledge few studies have addressed this issue. The key underlying factor between type of pregnancy and CHD-SGA being the placenta which could have a direct or indirect role in this association [20,67,68,69]. Jones et al. found a physiopathological explanation of SGA in HLHS based on placental histological analysis, a finding corroborated by other authors specializing in placentology rather than our study question [20]. For example, Matthiesen et al. investigated fetal and placental growth using Z scores [70]. Despite finding a slight difference in placental growth for HLHS, Matthiesen et al. observed an association between suboptimal placental weight and impaired fetal growth for TOF, VSD, and double outlet right ventricle [70]. Consequently, they concluded that placental growth is part of the causal pathway of the association between SGA and certain CHD. In conclusion, from our findings and based the literature, we hypothesize that both placental dysmorphology and abnormal fetal hemodynamics could play a role in the association between CHD and SGA. However further study is required to fully investigate this hypothesis.

This systematic review also confirmed ambiguity in the use of FGR and SGA in the literature. Despite the fact that SGA and FGR are quite distinct concepts, the terms were used interchangeably by different authors using a variety of definitions, cutoff thresholds and reference populations to infer the same meaning; SGA often being used as a proxy for FGR. A recent consensus based definition using a Delphi procedure defined FGR using exclusively ultrasound measurements [71]. While an international meeting of experts in 2007 reached a consensus on SGA, defining it as “a weight and/or length less than minus 2 standard deviations from the mean”; confusion still reigns [72,73]. Once our literature review was completed, we found an article that used the term “growth restriction in the newborn (GRN)” aimed at clarifying the situation [74]. This consensus-based definition, defined GRN as “birthweight < 3rd percentile compared to population or customized charts”. Alternatively, the presence of three out of the following five criteria: “birthweight <10th percentile compared to population or customized references, head circumference <10th percentile, length <10th percentile, prenatal diagnosis of FGR, and data on maternal pregnancy pathology” [48]. Of the 38 studies included in our systematic review, seven (18.4%) studies used a definition of SGA as birthweight < 3rd percentile thereby conforming to the recent definition of GRN. Although only two studies could be used in the meta-analysis, the proportion of GRN in all CHD was 6% (95% CI 6%–7%) [26,39,74]. However, we were unable to compare this to the proportion of GRN in the general population from the literature as this is a new concept. For the same reason our search did not find any study on CHD that specifically used the term GRN and further studies on this subject is required.

### 4.2. Strengths

Strengths of this systematic review are that a thorough search of the literature was carried out by a multidisciplinary team with specializations in pediatric cardiology, obstetrics, epidemiology, and library science. Following good research practice, the study protocol was registered in the PROSPERO database. The abstracts and articles were reviewed by two independent reviewers and data extraction followed standardized procedures. We evaluated the risk of bias using a validated standardized checklist. The set of studies included in the systematic review and particularly in the meta-analysis included many large population-based studies, which strengthened the external validity of the study in high resource countries. Results highlighted differences in the risk of SGA across different CHD subgroups, which can be useful for risk assessment and for generating hypotheses about the relation between CHD and growth restriction.

### 4.3. Limitations

Our study has certain limitations and caveats. Differences in practices and policies for prenatal diagnosis and termination of pregnancy for fetal anomaly (TOPFA) across populations and over time can result in changes in the proportion of SGA among newborns with CHD. As TOPFA concerns more severe CHD, all else equal, increases in TOPFA is likely to decrease the proportion of SGA among newborns with CHD. This is more likely to be the case for CHD associated with genetic or other severe anomalies.

The long period of time (1972–2018) for the publications included in the review could have affected the results, in part due to TOPFA but also changes in diagnosis of CHD and the and reference populations used for SGA. However, ⅔ of studies were published after 2009 and the meta-analysis results were often comparable for older and more recent studies.

The paucity of data on isolated subgroups of CHD complicated the interpretation of differences in the proportion of SGA across subgroups of CHD. In addition, the use of large and administrative databases in a number of studies could have been a source of inaccuracies because of coding and data entry errors.

As the majority of studies were from high resource, Western countries, (over two thirds of studies came from the USA), the results may not be generalizable to middle- and low-resource countries.

Finally, we did not evaluate publication bias due to the nature of the research question. Publication bias occurs when negative findings are less likely to be published and can be measured via visual inspection of funnel plots and Egger’s test. However, because there are no negative results in a prevalence study, we deemed these methods inappropriate for our meta-analysis [75].

## 5. Conclusions

Overall, the proportion of SGA in all CHD (20%) was 2-fold higher whereas that of isolated CHD (14%) was as 1.4-fold higher than the expected proportion in the general population. Although the available data have important limits, differences in the proportion of SGA for different subtypes of CHD suggest that there are different pathophysiological mechanisms underlying the relation between CHD and growth restriction. Further studies are required to disentangle the mechanisms of the association between CHD and growth restriction and the risks associated with growth restriction for newborns with CHD.

## Figures and Tables

**Figure 1 ijerph-17-03056-f001:**
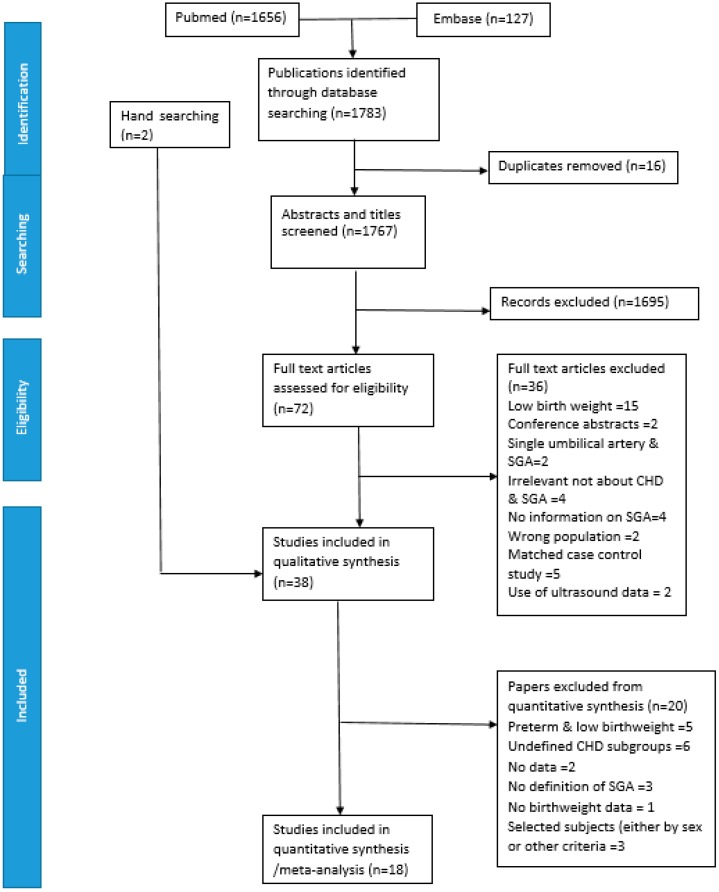
Flow chart to indicate the selection of studies.

**Figure 2 ijerph-17-03056-f002:**
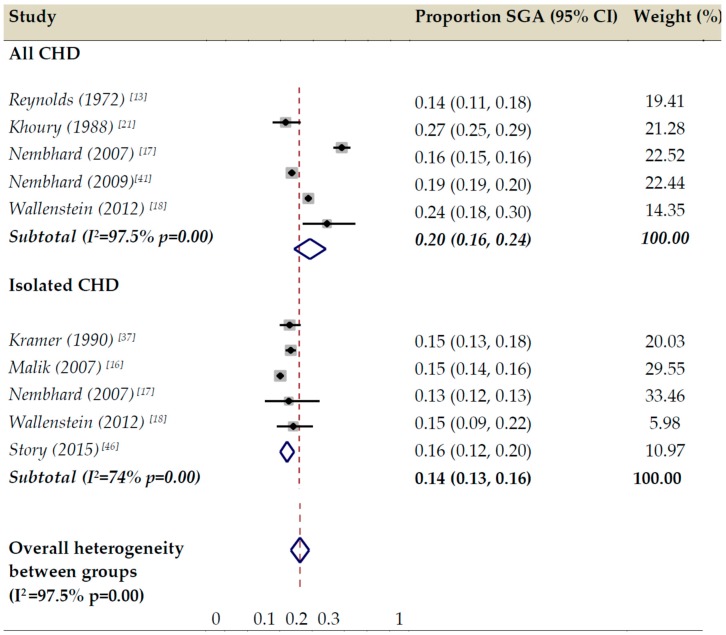
Forest plot of the meta-analysis of proportions of small for gestational age (SGA) in all and isolated congenital heart defects (CHD) according to 10th percentile cutoff threshold.

**Table 1 ijerph-17-03056-t001:** Number of citations according to different study characteristics.

**Characteristics of Study**	**Number of Publications**	**Number of Publications in MA**
**Year of Publication (*n* = 38)**		(*n* = 18)
1970–1979	3 (7.9%)	2 (11.1%)
1980–1989	1 (2.6%)	1(5.6%)
1990–1999	3 (7.9%)	2 (11.1%)
2000–2009	8 (21.1%)	6 (33.3%)
2010–2019	23 (60.5%)	7 (38.9%)
**Country (*n* = 38)**		(*n* = 18)
USA	26 (68.5%)	14 (77.8%)
Sweden	4 (10.5%)	1 (5.6%)
China	3 (8%)	1 (5.6%)
Italy	1 (2.6%)	0
France	1 (2.6%)	0
Chili	1 (2.6%)	0
UK	1 (2.6%)	1 (5.6%)
**Definition of SGA according to percentile (*n* = 38)**		(*n* = 18)
10th percentile (consensus definition of SGA)	22 (57.9%)	14 (77.8%)
3rd percentile	7 (18.4%)	4 (22.2%)
Undefined percentile	9 (23.7%)	0
**Consensus definition of SGA: 10th percentile: (*n* = 38)**		(*n* = 14)
No comparison	6 (27.2%)	4 (28.6%)
According to gestational age and sex	6 (27.3%)	4 (28.6%)
According to gestational age	4 (18.2%)	3 (21.4%)
According to gestational age, sex and race	3 (13.7%)	1 (7.1%)
According to gestational age and race	2 (9.1%)	2 (14.3%)
According to gestational age, race, sex, and single or multiple gestation	1 (4.6%)	0
Birthweight data provided for SGA	35 (92.1%)	18 (100%)
**Characteristics of Study**	**Number of Publications**	**Number of Publications in MA**
**SGA 1st aim of study**	17 (44.7%)	13 (72.2%)
**CHD**		
All	23	8
Isolated	10	7
**CHD subtype**		
HLHS	10	8
ToF	10	7
CoAo	8	7
TGV	7	7
AVSD	7	7
ASD	7	6
TA	3	3
CAT	3	3

Legend: MA—meta-analysis; SGA—small for gestational age; CHD—congenital heart defect; HLHS—hypoplastic left heart syndrome; ToF—Tetralogy of Fallot; VSD—ventricular septal defect; CoAo—coarctation of the aorta; TGV—transposition of great vessels; AVSD—atrioventricular septal defect; ASD—atrial septal defect; TA—tricuspid atresia; CAT—common truncus arteriosus.

**Table 3 ijerph-17-03056-t003:** Meta-analysis of proportions of SGA in different CHD subgroups (including genetic anomalies/syndromes) using the 10th percentile cutoff threshold.

Subgroup	Author	Pooled Proportion (95% CI)		% Weight
**HLHS**				
**Total pooled result**		21	(19–23)	
	Khoury (1988) [12]	23	(15–33)	7.36
	Nembhard (2009) [41]	23	(18–28)	22.81
	Williams (2010) [49]	20	(17–24)	48.79
	Swenson (2012) [47]	19	(15–24)	21.04
**ToF**				
**Total pooled result**		30	(24–37)	
	Khoury (1988) [12]	34	(25–43)	29.05
	Nembhard (2009) [41]	26	(23–30)	48.18
	Swenson (2012) [47]	36	(25–48)	22.77
**TGV**				
**Total pooled result**		17	(13–22)	
	Khoury (1988) [12]	17	(11–23)	28.79
	Nembhard (2009) [41]	20	(17–24)	41.34
	Swenson (2012) [47]	13	(8–18)	29.87
**VSD**				
**Total pooled result**		19	(18–20)	
	Khoury (1988) [12]	27	(24–31)	13.1
	Nembhard (2009) [41]	17	(16–19)	86.9
**CoAo**				
**Total pooled result**		22	(19–25)	
	Khoury (1988) [12]	28	(21–36)	19.06
	Nembhard (2009) [41]	20	(17–24)	80.94
**AVSD**				
**Total pooled result**		27	(21–32)	
	Khoury (1988) [12]	28	(20–38)	37.3
	Williams (2010) [49]	25	(18–33)	53.51
	Swenson (2012) [47]	32	(15–54)	9.19
**TA**				
**Total pooled result**		27	(21–35)	
	Williams (2010) [49]	30	(22–39)	74.84
	Swenson (2012) [47]	21	(10–37)	25.16
**CAT**				
**Total pooled result**		23	(17–30)	
	Khoury (1988) [12]	24	(11–41)	19.66
	Nembhard (2009) [41]	25	(17–34)	64.1
	Swenson (2012) [47]	18	(6–37)	16.24

**Legend:** HLHS—hypoplastic left heart syndrome; ToF—Tetralogy of Fallot; VSD—ventricular septal defect; CoAo—coarctation of the aorta; TGV—transposition of great vessels; AVSD—atrioventricular septal defect; TA—tricuspid atresia; CAT—common truncus arteriosus.

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
