# Peer review of "Children Born with Congenital Heart Defects and Growth Restriction at Birth: A Systematic Review and Meta-Analysis"

_ijerph, 2020, doi:10.3390/ijerph17093056_

Round 1
Reviewer 1 Report
The authors have addressed a difficult topic supported by a relatively large medical littérature, but qualitatively heterogeneous. The meta-analysis has been conducted professionnally and the study is in general well presented and discussed. I have only a couple of relatively minor points to consider for discussion:
-Endpoint selected is SGA (small for gestationnal age) which is sound for epidemiology but in clinical practice issue of gestational age at birth may be as important as SGA for prognosis and care. Both parameters determine birth weight (see for example Wogu, et al. Mediation analysis of gestational age, congenital heart defects, and infant birth-weight. BMC Res Notes 7, 926 (2014). The issue can be clarified for clinically-oriented readers.
- Only a fraction (50%) of the studies presented in table 1 are included in the meta-analysis. Accordingly, the table is poorly informative for characterizing input material in the analysis.
Other point :
Correct typing eror in sentence line 50-51.
Author Response
Reviewer 1
The authors have addressed a difficult topic supported by a relatively large medical littérature, but qualitatively heterogeneous. The meta-analysis has been conducted professionnally and the study is in general well presented and discussed. I have only a couple of relatively minor points to consider for discussion:
-Endpoint selected is SGA (small for gestationnal age) which is sound for epidemiology but in clinical practice issue of gestational age at birth may be as important as SGA for prognosis and care. Both parameters determine birth weight (see for example Wogu, et al. Mediation analysis of gestational age, congenital heart defects, and infant birth-weight. BMC Res Notes 7, 926 (2014). The issue can be clarified for clinically-oriented readers.
Response:
Our reviewer raises an interesting and complex question that is beyond the scope of our review. As cited in the Introduction of our paper, growth restriction,including its imperfect proxy SGA, is a known risk factor for short and long-term adverse health outcomes. However, as our reviewer points out, the relations between preterm delivery (gestational age in general), growth restriction, CHD and advere outcomes are complex and require both adequate data and appropriate modelling strategies for their relations to be sorted out.
Our ambition in this study was not to sort out these issues. Morever, there are not adequate data to address these issues by a systematic review of the literautre. However, in a study using our own data on a cohort of newborns with CHD, we are planning a path analysis approach to attempt to address some of the questions raised by our reviewer.
- Only a fraction (50%) of the studies presented in table 1 are included in the meta-analysis. Accordingly, the table is poorly informative for characterizing input material in the analysis.
Response: Table 1 has been corrected with a new column added detailing studies used in the meta-analysis. This complements the information provided in table 2 which details each individual study (*denoting studies used in the analysis) and figure 1(flow chart illustrating how studies used in the analysis were selected).
Other point :
Correct typing eror in sentence line 50-51.:
Response: correction made.
Thank you for your comments
Reviewer 2 Report
Ghanchi et al present a meta analysis of the relation between CHD and growth restriction at birth estimated by babies being small for gestational age. This is a well written manuscript with good fluency and progression. The discussion section on pathophysiological mechanisms and theories pertaining this relation of CHD and SGA as well as limitations section are excellent and presented well. Here are my minor recommendations:
First sentence of introduction needs a reference.
I appreciate that the authors mentioned that small for gestational age is an imperfect proxy for growth restriction in utero and at birth.
Line 51, delete the first word 'not'.
Line 91, definition of SGA needs at least one reference.
Thank you.
Author Response
Reviewer 2
Ghanchi et al present a meta analysis of the relation between CHD and growth restriction at birth estimated by babies being small for gestational age. This is a well written manuscript with good fluency and progression. The discussion section on pathophysiological mechanisms and theories pertaining this relation of CHD and SGA as well as limitations section are excellent and presented well. Here are my minor recommendations:
First sentence of introduction needs a reference.
Response: reference added (van der Linde, D., Konings, E. E., Slager, M. A., Witsenburg, M., Helbing, W. A., Takkenberg, J. J., & Roos-Hesselink, J. W. (2011). Birth prevalence of congenital heart disease worldwide: a systematic review and meta-analysis. Journal of the American College of Cardiology, 58(21), 2241-2247)
I appreciate that the authors mentioned that small for gestational age is an imperfect proxy for growth restriction in utero and at birth.
Line 51, delete the first word 'not'.
Response: correction made
Line 91, definition of SGA needs at least one reference.
- Response: reference added (Ego, A. (2013). Definitions: small for gestational age and intrauterine growth retardation. Journal de gynecologie, obstetrique et biologie de la reproduction, 42(8), 872-894)
Thank you.
Thank you for your comments